# Interactive Effects of Temperature and Plant Host on the Development Parameters of *Spodoptera exigua* (Hübner) (Lepidoptera: Noctuidae)

**DOI:** 10.3390/insects13080747

**Published:** 2022-08-19

**Authors:** Rameswor Maharjan, Jeongjoon Ahn, Hwijong Yi

**Affiliations:** 1Crop Production Technology Research Division, Department of Southern Area Crop Science, National Institute of Crop Science, Rural Development Administration, Miryang 50424, Korea; 2Research Institute of Climate Change and Agriculture, National Institute of Crop Science, Rural Development Administration, Ayeonno 1285, Jeju 63240, Korea

**Keywords:** beet armyworm, empirical models, population dynamics, forecasting, management

## Abstract

**Simple Summary:**

The beet armyworm *Spodoptera exigua* (Hübner) (Lepidoptera: Noctuidae) is an economic pest that causes significant damage to various vegetables and field crops in Korea. Timely forecasting of pest emergence and occurrence plays a vital role in pest management. To predict the spring emergence and population dynamics of *S. exigua*, constant temperature-mediated developmental parameters of *S. exigua* were studied with linear and nonlinear models on different plant hosts. The temperature and plant host significantly influenced the development of *S. exigua*. The total developmental time decreased with the increase in temperature, although there was no development at extreme temperatures. We developed a forecasting model, and our findings predicted the spring emergence of *S. exigua* in June. Thus, continuous monitoring of *S. exigua* in crop fields is needed to be ready to take advanced management measures against *S. exigua*.

**Abstract:**

This study investigated the effects of different temperatures (15, 20, 25, 27, 30, 35, and 40 °C) on the development rate of *Spodoptera exigua* (Hübner) eggs, larvae, pupae, and total immatures on plant hosts (soybean, maize, potato, and green pea). The eggs of *S. exigua* developed successfully at all the tested temperatures, except at 40 °C. The total developmental time (egg-adult) decreased with an increasing temperature from 15 to 35 °C on plant hosts. Stage-specific parameters such as the lower threshold temperature (TH) were determined using linear and nonlinear models (Sharpe-Schoolfield-Ikemoto [SSI]). The lower developmental threshold (LDT) and thermal constant (K) were determined using a linear model. The LDT and K for the total immature stage had respective values of 11.9 °C and 397.27° -day (DD) on soybean, 11.6 °C and 458.34° -day (DD) on maize, 11.2 °C and 446.23° -day (DD) on potato, 10.7 °C and 439.75° -day (DD) on green pea, and 12.2 °C and 355.82° -day (DD) on the artificial diet. The emergence frequency of adult *S. exigua* over the full range of constant temperatures was simulated using nonlinear developmental rate functions and the Weibull function. This study predicted the spring emergence date in the first to second weeks of June, with approximately five generations for plant hosts. The interaction of temperature and plant host also influenced the development and longevity of the adults. Overall, the findings of this study may be useful for predicting the number of generations, occurrence, population dynamics in crop fields, and management of *S. exigua*.

## 1. Introduction

*Spodoptera exigua* (Hübner), which is native to Southeast Asia, is a serious pest to numerous field crops, vegetables, and ornamentals including corn, soybean, potato, green pea, cotton, onion, peanut, and tomato [1,2,3,4,5]. It is a cosmopolitan species that feeds on 170 species, including 35 families of plants [6,7,8]. *Spodoptera exigua* generally has five, sometimes six instars of larvae [9]; starting from the fourth instar, larvae cause huge damage by consuming 80–90% of total food [10,11]. Chemical insecticides are commonly used to manage *S. exigua.* However, management of this pest has failed due to its rapidity in developing resistance to conventional chemical insecticides due to its wide host range, higher mobility and higher reproduction capacity [9,12], and a reduction in the haphazard use of broad-spectrum insecticides due to the substantial environmental concerns involved [13,14,15,16,17,18].

The host range of *S. exigua* is wide, and it can successfully breed on several plant hosts [1,6,11]. It has been reported that the oviposition and development of *S. exigua* can be affected by the physical/chemical attributes of the hosts as well as abiotic factors (temperature, humidity, and light) [6,11,19,20,21,22]. The rate of population build-up of *S. exigua* is responsible for the severity of the damage to the hosts, and abiotic factors such as temperature, humidity, and the nutritional quality of the hosts can play a role in population growth and damage intensity [3,11,22]. Consequently, there is a need for the development of environmentally sound alternative control measures for the integrated management of *S. exigua*.

It is important to construct a precise predictive model for adult emergence, and a prediction strategy that can serve as a critical component of an integrated pest management system wherein it facilitates decision making and enhances control efficacy [1,23]. Predicting the accurate seasonal occurrence of agricultural insect pests including *S. exigua* is significant for scheduling, sampling, and the selection of control tactics. Climatic factors in general have been shown to play significant roles in insect life; among such climatic factors, temperature has the greatest influence on population dynamics and the timing of biological events of insect species [24,25]. The development of insects occurs within narrow temperature ranges that vary between different insect species [26], and is sensitive to temperature changes [26,27,28,29,30,31,32,33]. Even slight alterations in temperature could cause spatial and temporal changes in the phenology of insects. The thermal requirements of the development of insects and biological agents are often used to predict their activity, seasonality, and population build-up [34,35,36]. The accurate and timely prediction of the occurrence of insect pests is one of the crucial aspects of an integrated pest management system, and it can be a foundation for understanding the sources and dynamics of local insect populations driven by temperature.

Temperature-dependent developmental models are essential for forecasting pest emergence [34], and there are several models based on the simplified analytic method [37,38,39,40,41,42] and biophysical approaches [43,44,45,46] that are often used to elucidate the relationship between temperature and the development rate. Although there is some information in the existing literature on the temperature-mediated development of *S. exigua* on artificial diets in Korea [19,21], there have been no studies examining the effects of temperature on the developmental biology and developmental distribution of *S. exigua* on different crop hosts in Korea. Although outbreaks of *S. exigua* are sporadic, they are very difficult to control with insecticides due to the rapid development of final instar larvae and insecticide resistance; therefore, early forecasting of the spring emergence of *S. exigua* is a crucial aspect of an effective intervention. Therefore, in this study, we examined the effects of constant temperatures on the development, adult longevity, and sex ratio relative to plant hosts soybean (*Glycine max* [L.] Merr.; cv. Daewon; Fabaceae), maize (*Zea mays* L.; cv. Hangkeummaschal; Poaceae), potato (*Solanum tuberosum* L.; cv. Sumi; Solanaceae), and green pea (*Pisum sativum* L.; cv. Spakal; Fabaceae), in order to develop empirical developmental models to estimate the thermal requirements of the development of *S. exigua*.

## 2. Materials and Methods

### 2.1. Insect Rearing

For the experiment, larvae of *S. exigua* were collected manually from soybean and potato fields at the Department of Southern Area Crop Science, National Institute of Crop Science (NICS), Rural Development Administration (RDA) (http://www.nics.go.kr/english/index.do, accessed on 1 January 2022), Miryang (Gyeongsangnam Province; 35°49′40′′ N, 128°74′01′′ E), Korea, in 2020. The field-collected larvae were maintained as separate colonies in the laboratory on an artificial diet [47]. To maintain the insect colonies, once the adults emerged, they were separately reared inside acryl cages (40 × 40 × 40 cm, with side ventilation) along with soybean plants (cv. Poonsannamool; greenhouse grown with commercial soil media, Baroker, Seoul Bio., Seoul, Republic of Korea), a honey solution (10%), and water. In the laboratory condition, all the cages were kept at 26 ± 1 °C, 60 ± 5% RH, and a 16:8 h L:D photoperiod. For the experiment, egg cluster layers attached on the soybean leaves collected by clipping the mother plant, and leaves with eggs clusters were used in the test. 

### 2.2. Plant Hosts

The plant hosts in this study were soybean (*Glycine max* [L.] Merr.; cv. Daewon; Fabaceae), maize (*Zea mays* L.; cv. Hangkeummaschal; Poaceae), potato (*Solanum tuberosum* L.; cv. Sumi; Solanaceae), and green pea (*Pisum sativum* L.; cv. Spakal; Fabaceae). Briefly, these plant hosts were selected because they are important cash crops in Korea and for their known associations with *S. exigua*. Single seeds of soybean, maize, and green pea, and seed pieces (approx. size: 20–30 g with a sprout) of potato, were planted individually in plastic pots (18 cm dia. *×* 13 cm height) in commercial soil media (Baroker, Seoul Bio., Seoul, Korea) without any pesticides. The growth of each plant host was synchronized by varying the planting date according to the varietal characteristics. All the plants were grown in a greenhouse at 28 ± 1 °C, 60–80% RH, and in a natural photoperiod.

### 2.3. Experimental Procedure

For oviposition, soybean plants (cv. Poonsannamool; 10–12 leaves/plant; 2 weeks old; greenhouse grown with commercial soil media (Baroker, Seoul Bio., Seoul, Korea) without pesticides) were exposed for 24–48 h to the laboratory colony of *S. exigua* (2–4 days old). They were kept in acryl cages (40 × 40 × 40 cm, with side ventilation) along with a honey solution (10%) and water, and held at 26 ± 1 °C, 60 ± 5% RH, and a 16:8 (L:D) h photoperiod. After exposure to adult *S. exigua* for 2 days, eggs were traced and collected for the experiment. Next, soybean leaves with eggs were placed into a Petri-dish (9.8 cm dia. *×* 4 cm height) and then placed inside humidity chambers (27 × 20 × 17 cm) in which the relative humidity (RH) (70–75%) level was set using saturated salt (NaCl) solutions (Duksan Pure Chemicals, Ansan-si, Gyeonggi Province, Korea) as described in [48]. The humidity chambers along with the eggs were then kept inside incubators (Eyela, model-MTI-202B, Tokyo, Japan) and set at temperatures of 15, 20, 25, 27, 30, 35, and 40 °C. All chambers had a photoperiod of 16:8 (L:D) hours. To monitor the temperature and humidity inside the environmental chambers, data loggers (Huato Log-USB, Huato Electronic Co. Ltd., Baoan District, Shenzhen, China) were used; the data loggers were attached inside the walls of the humidity chambers. Once the eggs hatched, each larva (1st instar) was separated and kept individually in a Petri-dish (5 cm dia. *×* 1.5 cm height, with topside ventilation) with the leaves of each plant host, and were fed an artificial diet [47]. The leaves (from the middle of the plant) of each plant host were replaced with new ones based on dryness. To measure the duration of each developmental stage, observations were conducted every 12 h (morning and evening) until adult emergence or death for each development stage. Once the pupae developed, their sexes were determined based on the morphology of the pupae [49], and were kept individually in Petri-dishes (5 cm dia. *×* 1.5 cm height, with topside ventilation). To determine the adult longevity based on the respective temperatures used during their development, emerged adults were also kept in the same Petri-dish with a honey solution (10%) as a food source.

### 2.4. Developmental Distribution Model

The two-parameter Weibull distribution function was used to analyze the relationships between the cumulative proportion of each life stage and the developmental times [34,50], which helps to predict the minor data with the Weibull function (Equation (1).
(1)$$F\left(x\right)=1-\exp(-{\left(\frac{x}{\alpha }\right)}^{\beta })$$
where F(x) is the cumulative proportion at development time *x*, *α* is the scale parameter, and *β* is the parameter of the curve shape. 

The median development time (i.e., time to 50% cumulative frequency) was calculated as *α* × [–Ln (0.5)]^1/*β*^. To generate a temperature-independent distribution model for each stage, the cumulative frequency was plotted against the normalized time, which was calculated by dividing the development time by the median development time at each temperature (i.e., development days/median days). The normalized data were pooled across temperatures and fit to the Weibull model [34].

### 2.5. Developmental Rate Models

Linear and nonlinear functions were used to describe the relationship between developmental rates (1/developmental periods) and temperature. The lower developmental threshold (LDT, $-\frac{b}{a}$) and thermal constant (K, 1/a) were estimated using a linear function (y = aT + b: where y = developmental rate, and T = assessed temperature) [37], and the parameters of the linear equation were estimated using the TableCurve 2D program (1996).

Among the various nonlinear equations that have been proposed to describe the relationship between developmental rates and temperature [51,52,53,54,55,56,57], we selected the Sharpe-Schoolfield-Ikemoto (SSI) model (Equation (2)).
(2)$$r\left(T\right)=\frac{\rho_{\phi }\frac{T}{T_{\phi }}\;\exp\left[\frac{\Delta HA}{R}\;\left(\frac{1}{T_{\phi }}-\frac{1}{T}\right)\right]}{1+\exp\left[\frac{\Delta HL}{R}\;\left(\frac{1}{TL}-\frac{1}{T}\right)\right]+\exp[\frac{\Delta HH}{R}\left(\frac{1}{TH}-\frac{1}{T}\right)]}$$

The SSI model based on thermodynamics was proposed by [43] and subsequently modified by [44,58]. It presents temperature-dependent reaction rates of active temperature ranges for a theoretical rate-controlling enzyme. The parameters of the SSI model were estimated using an R script (2015) developed by [54]. In Equation (2), *r*(*T*) represents the developmental rate at the absolute temperature *T* (°K); Δ*HA*, Δ*HL*, and Δ*HH* are enthalpy changes (Jmol^−1^); *R* is the universal gas constant; $\rho_{\phi }$ is the developmental rate at $T_{\phi }$; *TL* and *TH* are the temperatures at which the rate-controlling enzyme has an equal probability of being active or inactive, depending on low- or high-temperature inactivation, respectively. $T_{\phi }$ is the intrinsic optimum temperature at which the species can optimize its fitness to the environment [43,58]. 

### 2.6. Simulation of Spodoptera exigua Adult Emergence

The adult emergence of *S. exigua* was simulated according to temperature (°C) and time (day) by incorporating two functions of *S. exigua* development: the Weibull function for the development distribution, and the Performance model for the temperature-dependent rate. The rate of daily emergence at a given temperature was determined by the Performance model, and the cumulative frequency of adults was calculated using the Weibull distribution model (Equation (3)):(3)$$F\left(x,\;T\right)=1-\exp[-\left(\frac{x\;r\left(T\right)}{\alpha }{)}^{\beta }\right]$$
where F(x, T) is the cumulative proportion of the emergence of *S. exigua* adults at time x and constant temperature T, *x* is time (day), *r*(*T*) is the development rate model, and α and *β* are parameters from the Weibull equation.

### 2.7. Nutrient Contents of Plant Hosts

To analyze the lipid content, we prepared a 2 g sample of ground leaves by automatically weighing the container containing the sample, and then placed the sample in a fat extraction thimble (BUCHI Extraction Thimbles 25 × 100 mm) while using an n-hexane solvent for an automatic maintenance extraction system (BUCHI Labotechnik, B-811, AG, Meierseggstrasse 40. Postfach 9230 Flawil, Switzerland). The reaction was performed at 105 °C for 2 h and 40 min. After being allowed to cool for 30 min. in the desiccator, the weight of the extracted fat was measured [59].

To analyze the protein content, the plant host leaves were dried with a hot air dryer (40 °C) to 12–13% moisture content, at which point they were ground (C/11/1, Glenmills, 1439 Middletown, PA, USA). Fifty mg of the ground leaves was then wrapped in lactic acid paper and analyzed using an automatic Dumas combustion analyzer (Rapid N Cube, Elementar Analysensysteme GmbH, Langenselbold 63505, Germany) [60].

To analyze the sugar contents (stachyose, raffinose, sucrose, glucose, galactose, and fructose), the ground leaf sample (1 g) was placed in a falcon tube with 10 mL of 70% EtOH and shaken for 3 h in a stirrer. Twenty-four hours after the addition of EtOH, it was filtered using a filter paper. The filtered solution and H_2_O were mixed in a 1:1 (*v*:*v*) ratio using a vortex mixer and then analyzed by HPLC (Ultimate 3000; Thermo Scientific, Waltham, MA, USA) in an autosampler vial [59].

### 2.8. Statistical Analysis

Regression analyses were used to model the temperature-dependent development of each *S. exigua* stage, and the model parameter values for the linear and nonlinear functions were estimated using the TableCurve 2D Automated Curve Fitting program [61]. The effects of temperature on adult longevity and the nutrient contents among plant hosts were analyzed using analysis of variance (ANOVA) PROC GLM. The differences in adult longevity between females and males were compared with a *t*-test. All analyses were performed using the SAS program [62].

## 3. Results

### 3.1. Developmental Period

*Spodoptera exigua* developed successfully on all plant hosts (soybean, maize, potato, and green pea) at all temperatures, however at 40 °C, no eggs hatched. The developmental time of *S. exigua* was inversely related to the temperature from 15 to 35 °C. The egg, larva, pupa, and total immature developmental times were significantly influenced by temperature on all plant hosts (Table 1). The mean time required for development from eggs to adult emergence on soybean, maize, potato, green pea, respectively, ranged from 123.27 days at 15 °C up to 17.86 days at 35 °C, 125.95 days at 15 °C up to 20.03 days at 35 °C, 114.90 days at 15 °C up to 20.28 days at 35 °C, 109.27 days at 15 °C up to 16.15 days at 35 °C, and 125.47 days at 15 °C up to 16.32 days at 35 °C.

### 3.2. Developmental Distribution Model

The Weibull model was applied to describe the stage-specific frequency distribution of *S. exigua* against the normalized time (day/median) on soybean, maize, potato, green pea (Table 2, Figure 1). The cumulative frequency distribution of the developmental times of *S. exigua* at each temperature on each plant host was found to be well described by the Weibull model (r^2^ > 0.93, *p* < 0.0001) (Table 2).

The linear regression model was fit to the development rate data in the mid-range on all plant hosts (Table 3, and Figure 2, Figure 3 and Figure 4). The LDT and K for each life stage of *S. exigua* were estimated using linear regression analysis. The LDT values of total immature were 11.88 °C on soybean, 11.55 °C on maize, 11.18 °C on potato, 10.69 °C on green pea, and 12.18 °C on the artificial diet. Based on the LTDs, the K values of total immature were 397.27 (DD) on soybean, 458.34 (DD) on maize, 446.23 (DD) on potato, 439.75 (DD) on green pea, and 355.82 (DD) on the artificial diet.

Table 4 presents the estimated parameters of the nonlinear function. For all life stages, the temperature-dependent pattern of the *S. exigua* development rate over the entire range showed a typical skewed bell shape that featured a sharp decline if the rates were at exceedingly high temperatures above the optimal temperature (Figure 2, Figure 3 and Figure 4). The SSI model provided a significant fit to the temperature-dependent rate of *S. exigua* development on all plant hosts (Table 4, and Figure 2, Figure 3 and Figure 4). The respective intrinsic optimum temperatures (T*opt*.) of the control enzyme for the larva, pupa, and total immature periods were 34.0, 33.8, and 33.8 °C on soybean, 33.5, 40.6, and 34.7 °C on maize, and 34.3, 33.2, and 34.5 °C on potato; those for pupae and total immatures were 33.8 and 49.1 °C on green pea, and for eggs, larvae, pupae, and total immatures, the values were 34.3, 43.6, 32.9, and 33.7 °C on the artificial diet.

Given a cohort of *S. exigua* eggs under constant temperatures, the daily frequency of adult emergence was predicted using the nonlinear function, and it is presented in relation to temperature (°C) and time (day) (Figure 5). The results predicted that, at the optimum temperature, the adult emergence of *S. exigua* would occur earlier in a much shorter time, while at both extreme ends of temperature, the adult emergence would suffer from an extended delay. For instance, from the cohort of *S. exigua* eggs, adult emergence would occur in 19–30 days at 35 °C, whereas it would take 90–97 days at 15 °C and 25–30 days at 29 °C on all plant hosts.

### 3.3. Simulation of Adult Emergence

Based on the biofix of 1 January, the voltinism of *S. exigua* ranged from 4.46 to 5.79 generations on plant hosts over four years (Table 5). The number of generations was higher on the artificial diet in 2021, whereas it was lower on maize in 2020. However, the differences among hosts were not significant (*p* > 0.05).

### 3.4. Combined Effects of Temperature and Host

The combined analysis results show that temperature as well as plant hosts and the artificial diet significantly influenced the development of the life stages (egg, larva, pupa, and total immature) of *S. exigua* (Table 6). A longer time was needed for the development of all stages at the lower temperature (15 °C) whereas a shorter time was needed for the development of all stages at the higher temperature (35 °C) on all the plant hosts. The interaction between the temperatures and plant hosts was also significant (*p* < 0.0001) (Table 6 and Figure 6).

### 3.5. Adult Longevity

The adult longevity of both female and male *S. exigua* was significantly influenced by temperature on all plant hosts (Table 7). The difference in longevity between females and males was also significantly influenced by temperature except at 20 and 35 °C on soybean, at 15, 20, 25, and 30 °C on maize, at 15 °C on potato, at 27 °C on green pea, and at 25 °C on the artificial diet. The sex ratios (proportion female) between moths that emerged from soybean, maize, potato, green pea and the artificial diet were almost 1:1 ratio in all those cases (*p* > 0.05). Meanwhile, the highest female ratio (0.70) was at 25 °C on potato, and the lowest female ratio (0.40) was at 15 °C on potato and at 25 °C on maize (Table 7).

### 3.6. Survivorship

Figure 7 shows the survival of each life stage at different temperatures. The results show that the survival of *S. exigua* increased with temperature between 15 and 35 °C. The higher survival rates on soybean were 100% at 25, 27, and 35 °C for eggs, 95.12% at 27 °C for larvae, 95.65% at 15 °C for pupae, and 85.36% at 27 °C for total immatures; on maize, they were 100% at 25, 27, and 30 °C for eggs, 95.34% at 30 °C for larvae, 97.43 at 35 °C for pupae, and 86.36% at 35 °C for total immatures; on potato, they were 97.95% at 25 °C for eggs, 91.83% at 27 °C for larvae, 100% at 25, 27, 30, and 35 °C for pupae, and 88.23% at 27 °C for total immatures; on green pea, they were 98.03% at 25 °C for eggs, 95.83% at 15 °C for larvae, 97.87% at 30 °C for pupae, and 90.19% at 25 °C for total immatures; and on the artificial diet, they were 100% at 20, 25, 27, and 30 °C for eggs, 100% at 30 °C for larvae, 100% at 15 °C for pupa, and 87.17% at 30 °C for total immatures. Meanwhile, the lowest larval survival rate of total immatures (45.94% at 15 °C) was measured on maize.

### 3.7. Nutrient Contents

The nutrient contents exhibited significant differences among plant hosts for lipid (F_3,8_ = 19.15, *p* = 0.0005), protein (F_3,8_ = 707.77, *p* < 0.0001), raffinose (F_1,4_ = 7.41, *p* = 0.0529), sucrose (F_3,8_ = 3914.46, *p* < 0.0001), glucose (F_3,8_ = 760.55, *p* < 0.0001), and fructose (F_3,8_ = 5.71, *p* = 0.0218). Higher levels of lipid (5.31%) and protein (35.94%) were found in green pea and soybean, respectively. Galactose was only detected in green pea. Finally, stachyose was not detected in any plant hosts (Table 8).

## 4. Discussion

Global warming mediated by anthropogenic activities is anticipated to raise the earth’s temperature by approximately 1.5–5.8 °C by the end of the century, which is expected to lead to serious challenges to pest management and food security [63,64,65]. Temperature is the abiotic factor that determines the development rate and population growth of almost all organisms [66,67]. Insects are exothermic organisms, and the temperature is an important determinant of the pre-adult development rates of insects [68,69]. Therefore, it is crucial to understand the relationship between temperature and rate of development, because temperature influences insect biology, distribution, abundance, and damaging behavior [70,71,72,73,74]. Information on the influence of a constant temperature on the thermal requirements of a given insect pest is significant for the formulation of an integrated pest management program [75]. We investigated the effects of a wide range of temperatures on *S. exigua* development, and we estimated important thermal requirement parameters to understand the biological processes by using developmental rate models (linear and nonlinear) on different crop hosts (soybean, maize, potato, and green pea). This study details the prediction of *S. exigua* population dynamics in soybean, maize, potato, and green pea fields in Korea. Temperature affects the developmental time of *S. exigua* as it does for several insect pests in crop fields, including *Riptortus pedestris* (Thunberg) (Hemiptera: Alydidae), *Halyomorpha halys* (Stål) (Heteroptera: Pentatomidae), and *Aphis glycines* (Matsumura) (Hemiptera: Aphididae) [76,77,78]. The development trend of *S. exigua* at seven constant temperatures revealed decreases in the developmental time from eggs to adults with increases in temperature on all crop hosts (Table 1). The developmental time (1.81, 2.08, 2.93, 3.84, 5.86, and 13.04 days for eggs, 8.92, 10.73, 13.21, 14.76, 29.77, and 70.83 days for larvae, and 5.56, 5.58, 7.06, 8.76, 14.90, and 41.58 days for pupae at 35, 30, 27, 25, 20, and 15 °C, respectively) estimated on the artificial diet in the present study is similar to the estimated time for *S. exigua* on other artificial diets [19,21]. However, [19] estimated a slightly higher number of days for the development at 15 and 20 °C than we estimated in the present work. The developmental time estimated in this study is different from that of a study estimated on an artificial formula [11], which reported values of 0.76, 0.93, 0.93, 1.04, and 3.20 days for eggs, 5.99, 6.91, 7.77, 10.34, and 17.55 days for larvae, and 3.12, 3.99, 5.64, 6.63, and 7.68 days for pupae at 35, 30, 27, 25, and 20 °C, respectively. These discrepancies may be due to the differences in the diet fed to larvae and its composition, along with factors other than temperature such as genetic makeup, geographic origin, relative humidity, and photoperiod. Among the temperatures, *S. exigua* could not complete its development at 40 °C on all the plant hosts due to the fact that all eggs died. This type of unsuccessful egg/pupa development has also been reported by [11,79]. This failure in the development of *S. exigua* at the highest temperature might be interpreted to mean that high heat induces rapid dehydration of eggs, and that physiological disorders are responsible for the abnormal development and death of eggs. Substantial inhibition and unsuccessful development of eggs at high temperature were also reported in other insect species such as Hawaiian flower thrips, *Thrips hawaiiensis* (Morgan) (Thysanoptera: Thripidae) [80]. Prolonged exposure to high temperatures has been shown to produce various metabolic disorders in insects that ultimately lead to death [81]. High and low temperature exposure can also inactivate enzymes blocking cell cycle development, thereby substantially narrowing the temperature range for embryonic development in insects compared to the range of thermal tolerance in adults [82], which would explain why extreme temperatures had such adverse effects on the development of *S. exigua*.

The results regarding the combined effects of temperature and plant hosts (Table 6 and Figure 6) clearly show that, just like temperature, plant hosts also influenced the developmental time of *S. exigua*. This result is similar to that reported by [3,8,11], who also reported that plant hosts significantly affected the development of *S. exigua*. However, the developmental time of each stage varied according to the plant host. The results show that the insect had the shortest (109.27 days) developmental time when feeding on green pea, followed by when feeding on potato (114.90 days), and the longest (125.95 days) when feeding on maize (Table 1). However, the results of the present study contradict those reported by [3] for *S. exigua*, who found a longer developmental time (28 days at 27 °C) on pea. This discrepancy in developmental time may be due to variability in either the nutritional quality, or the quantity of host plant species [83]. These differences may also be related to the availability of different primary and secondary biochemicals on different host plants or different plant parts consumed by the larvae [84]. A study by [85] examining the feeding ecology of several species of bugs reported negative aspects of food containing limited nutrient contents that are essential for the normal development of bugs; specifically, insects feeding on such poor nutrient sources tend to store important nutrient contents such as lipids rather than utilize them for normal development, thus resulting in a longer development time. This plant host-based developmental time (which is shorter on green pea and longer on maize) of *S. exigua* is well supported by our nutrient content analysis, which clearly showed variations in the nutrient contents among the host plant leaves (Table 8). The contents of all nutrients (lipid, protein, raffinose, sucrose, glucose, galactose, and fructose) except stachyose were detected in green pea, with a higher lipid content (5.31%) and protein content (35.94%) than maize (lipid: 4.74%; protein: 29.78%). In contrast to our study, [84] reported lower developmental times of 3 days for eggs, 14.91 and 13.10 days for larvae, 7.02 and 6.66 days for pupae, and 24.93 and 22.74 days for egg-adults at 26 ± 1 °C on maize (*Z. mays*) and soybean (*G. max*), respectively. The authors of [8] reported lower total developmental times of 16.91, 27.21, 41.63, and 120.50 days at 30, 25, 20, and 15 °C, respectively, on sugar beet. These differences may be attributed to food (sugar beet), geographic origin, genetic makeup, and the exposure and experience of differential adaption to the environmental regimes. This phenomenon (variations in developmental time based on the host plant and nutrient quality) of *S. exigua* has been reported in several previous studies [3,6,9,84,86,87,88,89,90].

Origination and the development-based lower temperature threshold can be used to estimate the population development of organisms [91]. The estimations of LTDs for different stages of *S. exigua* from the linear model estimated by [19,21] on the artificial diet as well as [8], are all in contrast to the findings of our study. The authors of [19,21] estimated higher LTDs for eggs, larvae, pupae and total immatures. However, [8] reported lower LDTs for the egg, larva, and pupa stages on sugar beet than we estimated on soybean and the artificial diet.

The authors of [11] also reported lower LDTs for eggs (7.50 °C), larvae (7.96 °C), pupae (5.60 °C), and total immatures (5.60 °C) than our estimates for eggs (12.80 °C), larvae (12.43 °C), pupae (13.11 °C), and total immatures (12.18 °C) on the artificial diet. According to the present study, eggs required 39.91 DD to hatch on the artificial diet, which is similar to the thermal constant (39.37 DD) reported by [9] and lower than the thermal constant (23.62) reported by [11]. However, Refs. [79,92] reported higher thermal constant values (42.55 and 49.15 DD, respectively) on an artificial diet than the present study. In contrast to our study, Ref. [8] reported a lower thermal constant (40.16 DD) for egg hatching on sugar beet than we estimated on the plant hosts. The thermal constant of the larval period in our study (195.24 DD) was higher on the artificial diet than that estimated by [21] (155.72 DD), [19] (155.80 DD), [9] (128.70 DD), and [11] (191.49 DD). Similarly, we also estimated a higher thermal constant on plant hosts than that reported by [8] on sugar beet (174.83 DD) and by [9] on Pigweed (123.20 DD) and cotton (157.70 DD). This variation may also be due to differences in the nutrient quality of plant hosts i.e., it is possible that a factor other than temperature affects the thermal requirement, as suggested by the significant differences seen among plant hosts, temperatures, and the interaction between plant host and temperature in this study (Table 6). With similar findings to this study, Ref. [93] reported that there is a significant interaction of temperature and diet that ultimately alters the developmental time of larvae of *S. exigua*, even at an identical temperature on different diets. This discrepancy could also be due to the differences in each study’s experimental methodology. For example, in this study, for pupal development, we used a Petri dish (5 cm dia. *×* 1.5 cm height, with topside ventilation) with the leaves of each plant host above a piece of tissue paper to provide a rough surface. Each Petri dish was then closed tightly with para film to protect the larvae from escaping and to maintain moisture. However, we did not provide soil for pupation, whereas Ref. [8] provided 1 cm height soil for pupation purposes. One should therefore consider the possibility that the materials and methodologies adopted could affect pupal development and its thermal constants. 

In existing research studies, several mathematical functions have been developed and employed to describe the insect development rates across thermal regimes. The selection of models is generally based on the choice of authors and is strongly subject to field -associated biases. While certain models consistently perform better within a specific context, there is no consensus on which are generally best across a wide range of applications [94]. Many investigations related to temperature-dependent development are carried out with a single model or a few models targeting a particular taxonomic group, often without justification. Some of the important qualities of the model, such as the potentially superior predictive power or other beneficial qualities may be overlooked [95]. To overcome this limitation, we selected several models and analyzed the observed data using different models. One nonlinear mathematical function (SSI) was used to adequately describe the developmental rate versus temperature curve, because the relationship between the developmental rate and temperature is curvilinear near extremes. Both the temperature threshold and the temperature maximum using the SSI nonlinear model were employed to estimate the temperature ranges for insect development, because the model provides clear biophysical meaning and thermodynamic information among model parameters [82,96]. Further, Ref. [97] suggested that the SSI function performed as well as, or relatively better than, other functions. The developmental rate of *S. exigua* was well fitted and described using a nonlinear model, which is indicated by the high coefficient (r^2^) values (0.98–0.99) and the minimum variance of the estimated parameters on all plant hosts. The intrinsic optimum temperature ($T_{\phi }$) value for development is the most critical factor that determines the fitness of an optimum life history strategy [58], which suggests the involvement of the maximal active state of enzymes in the development process [98,99]. The $T_{\phi }$ values in this study using the SSI model were 26.6 °C on soybean, 23.6 °C on maize, 22.3 °C on potato, 33.1 °C on green pea, and 26.7 °C on the artificial diet, which differ from those of [8], who estimated 28.5 °C on sugar beet. The T low values estimated by the SSI model varied between 12.11 °C and 14.91 °C, and the T high values estimated by the SSI model for the total immature stage were 35.6 °C on soybean, 37.1 °C on maize, 37.9 °C on potato, 50.3 °C on green pea, and 37.0 °C on the artificial diet (Table 4). The estimated T low and T high values in the present study differ from those reported by [8], who estimated respective values of 13.3 °C and 34.6 °C on sugar beet. The $T_{\phi }$ value (33.1 °C on green pea) estimated by the SSI model for the total immature stage refers to the temperature that can make the population obtain maximum fitness as the true optimal temperature, in which the developmental rate was 0.0544 day-1.

Standard laboratory tests under a standard set of conditions, such as constant temperature and other controlled and replicable conditions with a particular physiological or behavioral response or survival, often produce relatively simple relationships. However, when comparing laboratory conditions with field conditions, field conditions are much more complex and variable. and within this complexity, several types of known or unknown traits might be involved [100]. The authors of [101] reported an alteration in the phenotype of *Drosophila melanogaster* Meigen due to the effect of developmental and adult temperature acclimation. The authors of [100] reported adaptive effects of acclimation in both laboratory and field tests, with stronger effects in the field test. Though there is a difference in the relationship between the laboratory and field, the estimated laboratory-based traits might still be significant and highly relevant to field performance, but this critical assumption needs to be verified [102].

The results of the present study provide substantial evidence indicating that temperature and plant hosts significantly affect the longevity of *S. exigua* (Table 7). The longest longevity for *S. exigua* was found at 30 °C on soybean, while the shortest was found at 35 °C on potato. As stated above, longevity differences among plant hosts can also be related to the nutritional quality and quantity of plant host species as well as the primary and secondary biochemical contents available on those plant hosts [83,84]. Significant differences were also found between the longevity values of females and males. On all plant hosts, females had longer longevity than males. Several previous studies have also reported differences in longevity between sexes of *S. exigua* [19,21] and other insect species, such as the azuki bean weevil (*Callosobruchus chinensis*, Coleoptera: Bruchidae) [26] and the potato leafminer fly (*Liriomyza huidobrensis*, Diptera: Agromyzidae) [103]. These differences in adult longevity could be due to (i) complex interactions between the specific local environmental conditions and sex-specific costs of reproduction, (ii) epigenetic control of longevity by imprinting through DNA methylation, and (iii) increased fecundity and protection from aging stemming from the act of mating or components from the male ejaculate [104]. 

Linear models are widely used to estimate the lower temperature threshold and thermal constant of insect species [105], though researchers have highlighted many drawbacks [106,107]. Despite these shortcomings, linear models remain widely used because they require minimal data input for formulation with easy calculation. Therefore, their application has generally been found to yield correct values with negligible differences in accuracy [36]. For these reasons, we adopted and used a linear model to estimate the parameters of temperature-dependent development on different plant hosts and on an artificial diet. By using a linear model with 397.27 DD on soybean, 458.34 DD on maize, 446.23 DD on potato, 439.75 DD on green pea, and 355.82 DD on the artificial diet, we tested the simple application of a degree-day model with the biofix of 1 January [33,108] to predict the number of generations of *S. exigua*, which resulted in 5.2, 4.6, 5.0, 5.3, and 5.6 generations on soybean, maize, potato, green pea, respectively. The resulting spring emergence date of *S. exigua* was 2–8 June in 2018, 6–14 June in 2019, 4–9 June in 2020, and 6–13 June in 2021 on plant hosts in Korea (Table 5) (Maharjan, unpublished). As a result, this study provides important information on the temperature-dependent development of this polyphagous pest in Korea, which is expected to be useful for the prediction modeling of the distribution expansion and population regulation of *S. exigua* from a climate change perspective [107,109]. The model developed in the present study could contribute to the development of integrated pest management strategies including spray timing [110,111], even with limited capacity of extrapolation from the laboratory-based parameter estimation [112,113].

## Figures and Tables

**Figure 1 insects-13-00747-f001:**
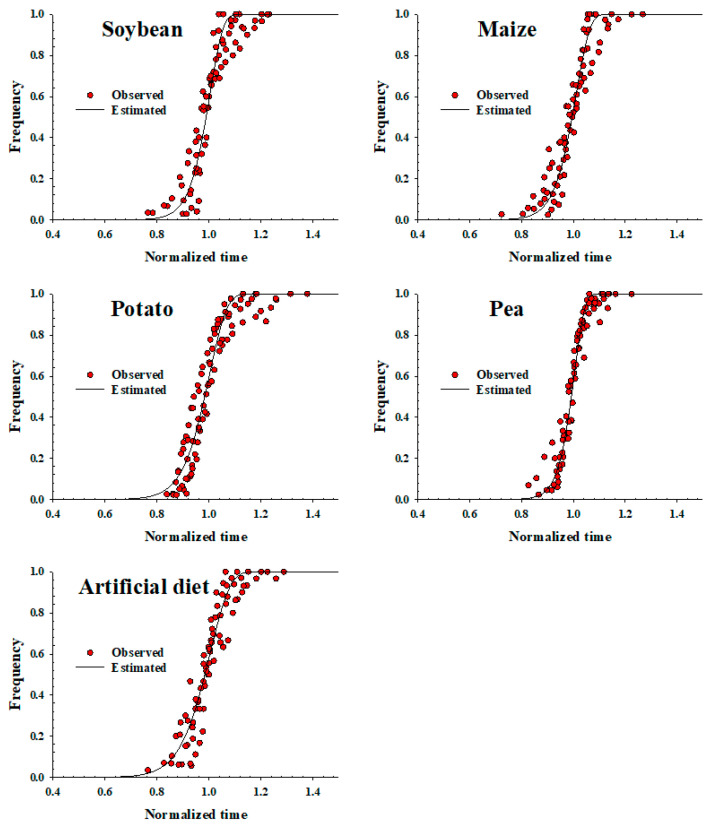
Cumulative frequency of the development of *Spodoptera exigua* using plant hosts (soybean, maize, potato and green pea) and the artificial diet as food sources against the normalized time (day/median day), fit to the Weibull function. Red circles: observed data; solid lines: estimated value.

**Figure 2 insects-13-00747-f002:**
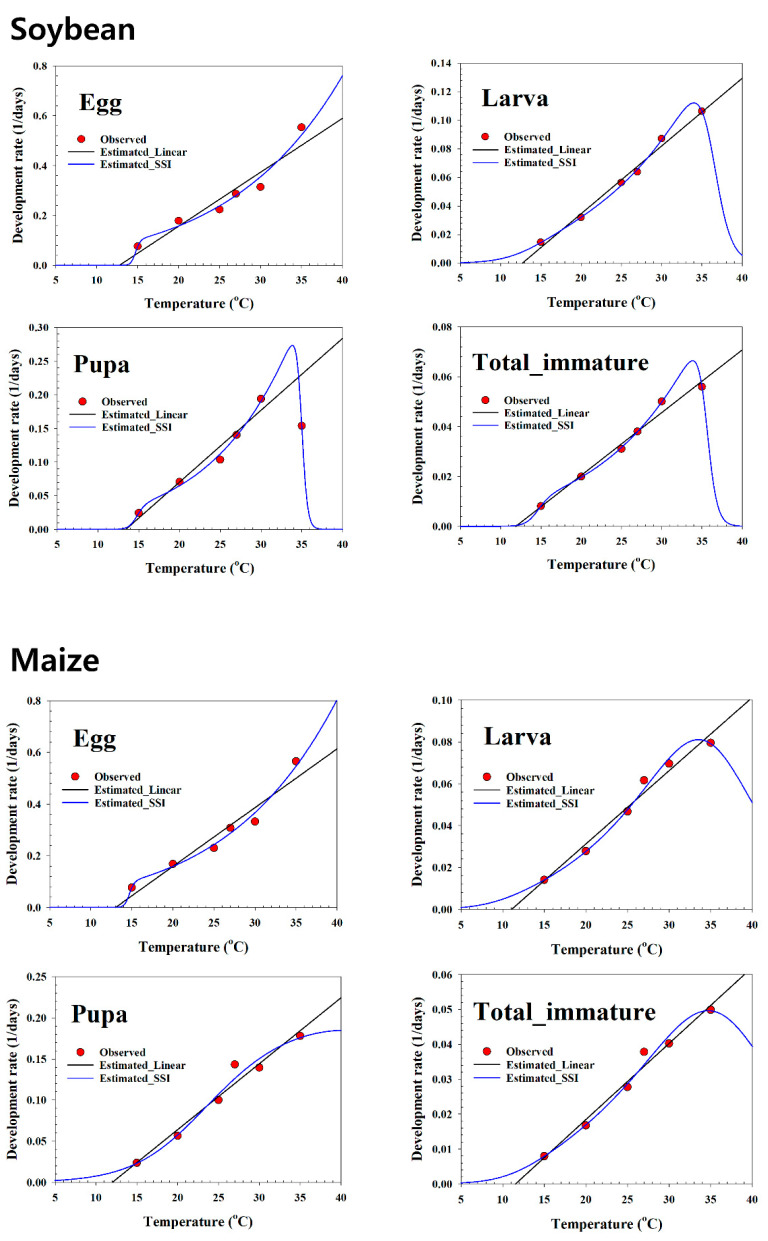
Linear and non-linear models for the temperature-dependent rate (1/day) of the development of *Spodoptera exigua* using soybean and maize as food sources. Red circles: observed data; solid blue lines: estimated nonlinear result; long solid black lines: estimated linear result.

**Figure 3 insects-13-00747-f003:**
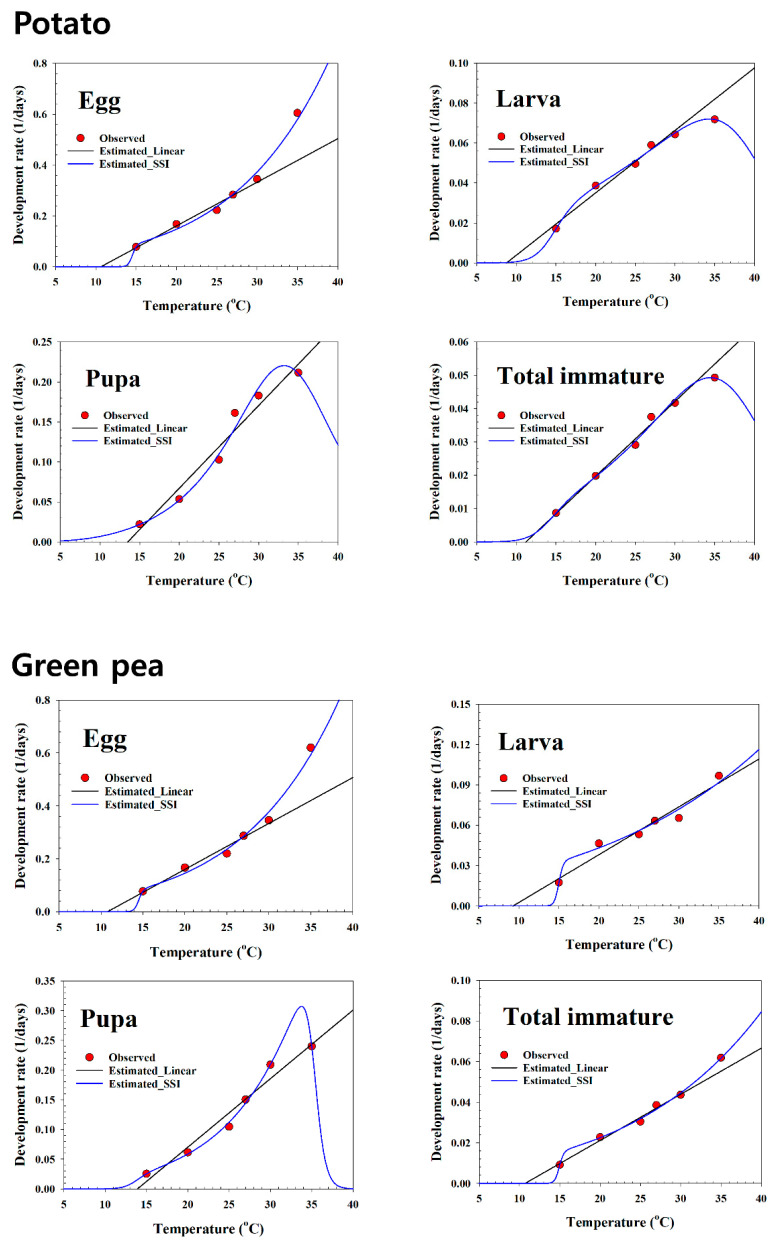
Linear and non-linear models for the temperature-dependent rate (1/day) of the development of *Spodoptera exigua* using potato and green pea as food sources. Red circles: observed data; solid blue lines: estimated nonlinear result; long solid black lines: estimated linear result.

**Figure 4 insects-13-00747-f004:**
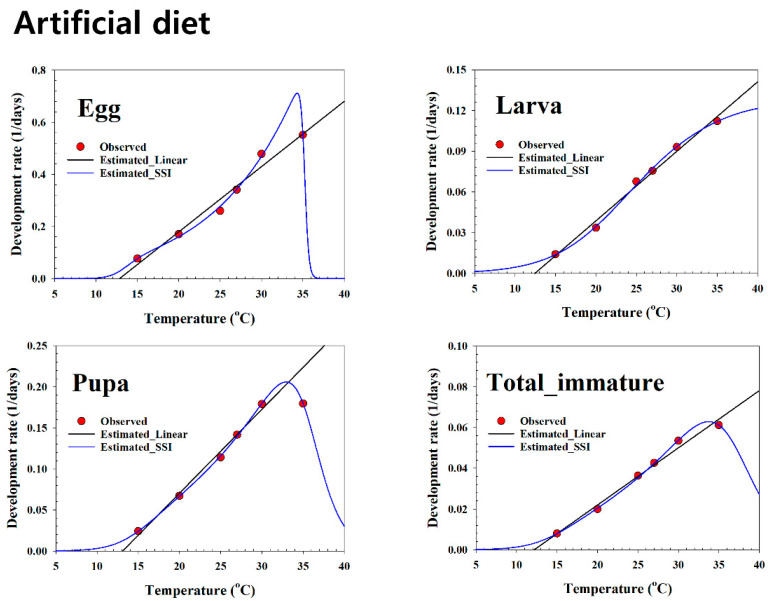
Linear and non-linear models for the temperature-dependent rate (1/day) of the development of *Spodoptera exigua* using the artificial diet as a food source. Red circles: observed data; solid blue lines: estimated nonlinear result; long solid black lines: estimated linear result.

**Figure 5 insects-13-00747-f005:**
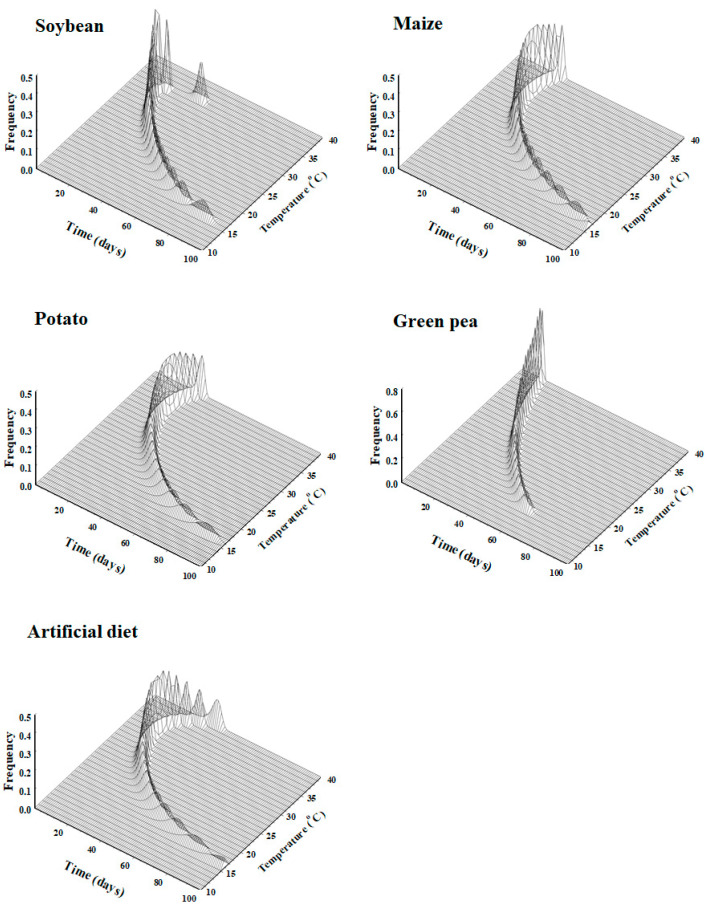
Simulation frequency of *Spodoptera exigua* adult emergence from an egg cohort in relation to time (day) and constant temperature (°C) on plant hosts (soybean, maize, potato and green pea) and on the artificial diet.

**Figure 6 insects-13-00747-f006:**
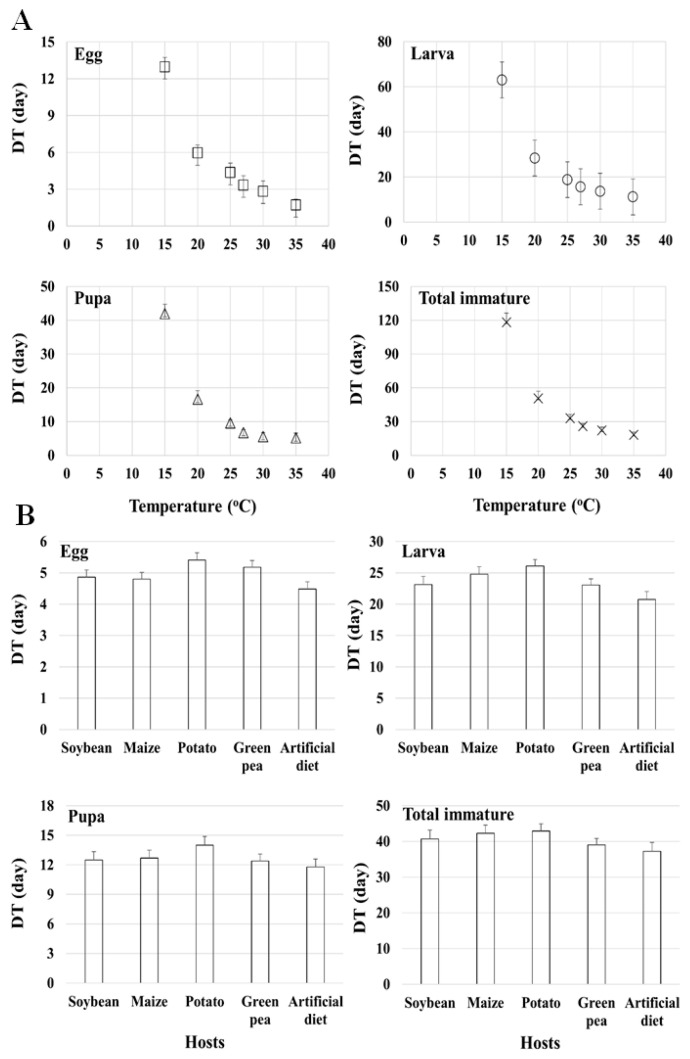
Combined effects of temperature (°C) (**A**) and plant hosts (**B**) on the developmental time (DT) of eggs, larvae, pupae, and total immatures of *Spodoptera exigua*.

**Figure 7 insects-13-00747-f007:**
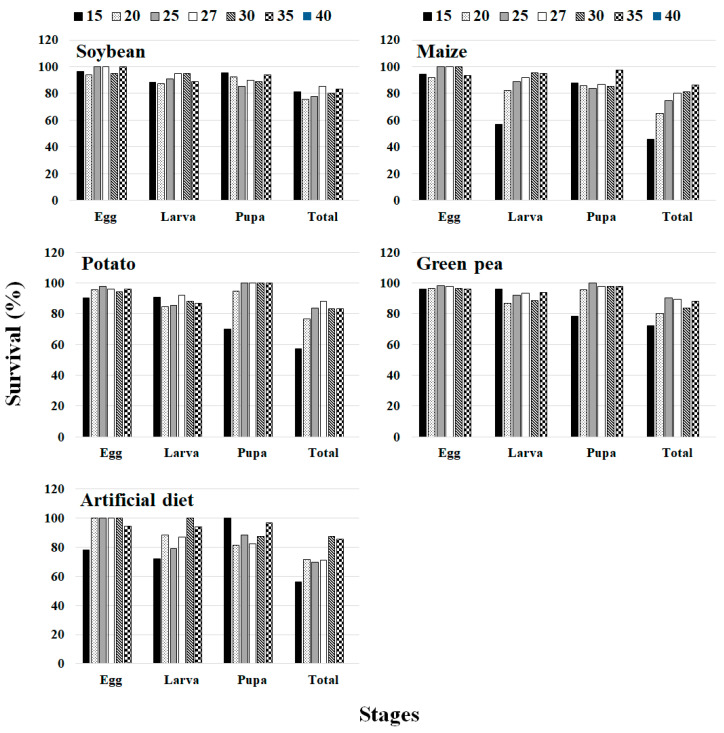
Survivorship of different stages (egg, larva, pupa, and total) of *Spodoptera exigua* at different temperatures (15, 20, 25, 27, 30, 35, and 40 °C) on plant hosts (soybean, maize, potato and green pea). At the temperature of 40 °C, no eggs hatched.

**Table 1 insects-13-00747-t001:** Development time in days (mean ± SE) for *Spodoptera exigua* on soybean, maize, potato, green pea and the artificial diet as food sources at a constant temperature.

Hosts	Stage	Temperature (°C)						
		40	35	30	27	25	20	15
Soybean	Egg	*ˠ*	1.80 ± 0.1f	3.18 ± 0.1e	3.48 ± 0.1d	4.47 ± 0.1c	5.93 ± 0.1b	13.13 ± 0.2a
	n	44	36	40	41	45	33	27
	Larva	-	9.40 ± 0.4f	11.48 ± 0.3e	15.66 ± 0.3d	17.76 ± 0.3c	33.25 ± 0.4b	69.13 ± 0.5a
	n	-	36	38	41	45	31	26
	Pupa	-	6.50 ± 0.3f	5.15 ± 0.2e	7.12 ± 0.2d	9.67 ± 0.2c	15.28 ± 0.2b	41.13 ± 0.4a
	n	-	32	36	39	41	27	23
	Total immature	-	17.86 ± 0.4f	19.95 ± 0.3e	26.27 ± 0.3d	32.22 ± 0.3c	54.00 ± 0.3b	123.27 ± 0.6a
	n	-	30	32	35	35	25	23
Maize	Egg	*ˠ*	1.76 ± 0.1e	3.01 ± 0.1d	3.25 ± 0.1d	4.34 ± 0.1c	5.94 ± 0.1b	13.01 ± 0.1a
	n	36	44	43	50	55	37	37
	Larva	-	12.57 ± 0.2f	14.36 ± 0.3e	16.22 ± 0.3d	21.38 ± 0.3c	35.96 ± 0.4b	71.02 ± 0.9a
	n	-	41	43	50	55	34	35
	Pupa	-	5.61 ± 0.2e	7.17 ± 0.2d	6.97 ± 0.2d	10.00 ± 0.1c	17.68 ± 0.2b	42.30 ± 0.2a
	n	-	39	41	46	49	28	25
	Total immature	-	20.03 ± 0.3f	24.84 ± 0.4e	26.43 ± 0.4d	36.02 ± 0.2c	59.72 ± 0.6b	125.95 ± 0.2a
	n		38	35	40	41	24	22
Potato	Egg	*ˠ*	1.65 ± 0.1f	2.90 ± 0.1e	3.53 ± 0.1d	4.89 ± 0.1c	5.96 ± 0.1b	12.82 ± 0.1a
	n	46	48	54	51	49	47	61
	Larva	-	13.91 ± 0.1f	15.54 ± 0.2e	16.94 ± 0.5d	20.15 ± 0.4c	25.82 ± 0.6b	58.31 ± 0.7a
	n	-	46	51	49	48	45	55
	Pupa	-	4.72 ± 0.1e	5.46 ± 0.1d	6.20 ± 0.1d	9.73 ± 0.1c	18.65 ± 0.2b	45.06 ± 0.3a
	n	-	40	46	45	41	38	37
	Total immature	-	20.28 ± 0.2d	23.97 ± 0.3d	26.64 ± 0.5c	34.34 ± 0.4c	50.63 ± 1.1b	114.90 ± 1.0a
	n	-	40	46	45	36	36	50
Green pea	Egg	*ˠ*	1.61 ± 0.1f	2.89 ± 0.1e	3.48 ± 0.1d	4.55 ± 0.1c	6.00 ± 0.1b	13.02 ± 0.1a
	n	51	51	55	47	51	55	50
	Larva	-	10.33 ± 0.1e	15.30 ± 0.1d	15.08 ± 0.2d	18.78 ± 0.2c	21.51 ± 0.2b	57.59 ± 0.4a
	n	-	49	53	46	50	53	48
	Pupa	-	4.16 ± 0.1e	4.78 ± 0.1e	6.65 ± 0.1d	9.55 ± 0.1c	16.29 ± 0.3b	39.30 ± 0.4a
	n	-	46	47	43	46	46	46
	Total immature	-	16.15 ± 0.1f	22.91 ± 0.1e	25.94 ± 0.2d	32.89 ± 0.2c	43.86 ± 0.3b	109.27 ± 0.7a
	n		45	46	42	46	44	34
Artificial diet	Egg	*ˠ*	1.81 ± 0.1e	2.08 ± 0.1e	2.93 ± 0.04d	3.84 ± 0.2c	5.86 ± 0.1b	13.04 ± 0.2a
	n	41	34	39	45	43	42	32
	Larva	-	8.92 ± 0.2f	10.73 ± 0.2e	13.21 ± 0.2d	14.76 ± 0.5c	29.77 ± 0.5b	70.83 ± 0.7a
	n	-	32	39	45	43	42	25
	Pupa	-	5.56 ± 0.2e	5.58 ± 0.1e	7.06 ± 0.1d	8.76 ± 0.2c	14.90 ± 0.4b	41.58 ± 1.1a
	n	-	30	39	39	34	37	11
	Total immature	-	16.32 ± 0.3f	18.67 ± 0.2e	23.45 ± 0.3d	27.48 ± 0.6c	50.06 ± 0.8b	125.47 ± 0.3a
	n	-	28	34	32	30	30	18

ˠ: no eggs hatched; *Soybean*: Egg- F_5, 34_ =625.56, *p* < 0.0001, Larva- F_5, 195_ =3359.47, *p* < 0.0001, Pupa-F_5, 1173_ =3344.73, *p* < 0.0001, Total immature-F_5, 142_ =12868.2, *p* < 0.0001; *Maize*: Egg-F_5, 243_ =1630.21, *p* < 0.0001, Larva-F_5, 222_ =2613.65, *p* < 0.0001, Pupa-F_5, 194_ =4841.45, *p* < 0.0001, Total immature-F_5, 194_ =5523.43, *p* < 0.0001; *Potato*: Egg-F_5, 290_ =1876.05, *p* < 0.0001, Larva-F_5, 125_ =1230.83, *p* < 0.0001, Pupa-F_5, 145_ =3290.22, *p* < 0.0001, Total immature-F_5, 65_ =914.74, *p* < 0.0001; *Green pea*: Egg-F_5, 291_ =1741.33, *p* < 0.0001, Larva-F_5, 265_ =6692.41, *p* < 0.0001, Pupa-F_5, 214_ = 5487.77, *p* < 0.0001; Total immature-F_5, 214_ =11617.9, *p* < 0.0001; *Artificial diet*: Egg-F_5, 219_ =841.70, *p* < 0.0001, Larva-F_5, 123_ =2218.33, *p* < 0.0001; Pupa-F_5, 167_ =1833.33, *p* < 0.0001, Total immature-F_5, 32_ =2194.47, *p* < 0.0001. Means followed by the same letters in a row are not significantly different among temperatures (ANOVA, Tukey’s HSD test, *p*< 0.05).

**Table 2 insects-13-00747-t002:** Parameter estimates (mean ± SE) of the Weibull distribution models for the development of *Spodoptera exigua* using plant hosts (soybean, maize, potato and green pea) and on the artificial diet as food sources against the normalized time (day/mean).

Hosts	Parameters	Estimate ± SE	r^2^
Soybean	α	1.0067 ± 0.0019	0.95
	β	21.0532 ± 1.3651	
Maize	α	1.0137 ± 0.0020	0.95
	β	19.9921 ± 1.1130	
Potato	α	1.0067 ± 0.0028	0.93
	β	16.0278 ± 0.9462	
Green pea	α	1.0062 ± 0.0012	0.97
	β	26.4978 ± 1.1775	
Artificial diet	α	1.0101 ± 0.0024	0.95
	β	14.2051 ± 0.7772	

**Table 3 insects-13-00747-t003:** Linear regression analysis for *Spodoptera exigua* using plant hosts (soybean, maize, potato and green pea) and the artificial diet as food resources.

Hosts	Life Stage	Linear Regression	LDT	K (DD)
Soybean	Egg	−0.2763 + 0.0216T	12.76	46.19
	Larva	−0.0602 + 0.0047T	12.69	210.78
	Pupa	−0.1432 + 0.0107T	13.42	93.71
	Total immature	−0.0299 + 0.0025T	11.88	397.27
Maize	Egg	−0.2959 + 0.0227T	13.01	43.98
	Larva	−0.0388 + 0.0035T	11.08	285.68
	Pupa	−0.0963 + 0.0080T	12.01	124.73
	Total immature	−0.0252 + 0.0022T	11.55	458.34
Potato	Egg	−0.1825 + 0.0172T	10.63	58.25
	Larva	−0.0272 + 0.0031T	8.74	320.35
	Pupa	−0.1394 + 0.0103T	13.49	96.74
	Total immature	−0.0251 + 0.0022T	11.18	446.23
Green pea	Egg	−0.1872 + 0.0174T	10.78	57.60
	Larva	−0.0329 + 0.0036T	9.28	281.26
	Pupa	−0.1603 + 0.0115T	13.90	86.72
	Total immature	−0.0243 + 0.0022T	10.69	439.75
Artificial diet	Egg	−0.3216 + 0.0251T	12.84	39.91
	Larva	−0.0636 + 0.0051T	12.43	195.24
	Pupa	−0.1340 + 0.0102T	13.11	97.84
	Total immature	−0.0342 + 0.0028T	12.18	355.82

LDT: lower developmental threshold. K: thermal constant; DD: degree-days. Soybean: Egg F_1, 4_ = 38.9968, *p* < 0.0033, r^2^ = 0.91, Larva F_1, 4_ = 348.169, *p* < 0.00005, r^2^ = 0.99, Pupa F_1, 4_ = 62.3627, *p* < 0.0042, r^2^ = 0.95, Total immature F_1, 4_ = 217.364, *p* < 0.0001, r^2^ = 0.98. Maize: Egg F_1, 4_ = 51.1423, *p* < 0.0020, r^2^ = 0.93, Larva F_1, 4_ = 155.836, *p* < 0.0002, r^2^ = 0.98, Pupa F_1, 4_ = 96.5481, *p* < 0.0006, r^2^ = 0.96, Total immature F_1, 4_ = 199.006, *p* < 0.0001, r^2^ = 0.98. Potato: Egg F_1, 3_ = 161.44, *p* < 0.0011, r^2^ = 0.98, Larva F_1, 3_ = 147.65, *p* < 0.0012, r^2^ = 0.98, Pupa F_1, 4_ = 88.30, *p* < 0.0007, r^2^ = 0.96, Total immature F_1, 3_ = 266.11, *p* < 0.0005, r^2^ = 0.99. Green pea: Egg F_1,3_ = 134.94, *p* < 0.0013, r^2^ = 0.98, Larva F_1, 4_ = 69.57, *p* < 0.0011, r^2^ = 0.95, Pupa F_1, 4_ = 99.91, *p* < 0.0005, r^2^ = 0.96, Total immature F_1, 3_ = 222.49, *p* < 0.0006, r^2^ = 0.99., Artificial diet: Egg F_1, 4_ = 123.226, *p* < 0.0003, r^2^ = 0.97, Larva F_1, 4_ = 427.002, *p* < 0.00003, r^2^ = 0.99, Pupa F_1, 3_ = 343.055, *p* < 0.0003, r^2^ = 0.99, Total immature F_1, 4_ = 314.577, *p* < 0.00006, r^2^ = 0.99.

**Table 4 insects-13-00747-t004:** Parameter estimates of the nonlinear developmental rate model for *Spodoptera exigua* using plant hosts (soybean, maize, potato and green pea) and the artificial diet as food resources.

Hosts	Function		Life Stage			
			Egg	Larva	Pupa	Total Immature
Soybean	SSI	*Ρ_Φ_*	0.4822	0.0828	0.1393	0.0367
		*Τ_Φ_*	307.0976	302.6459	300.184	299.7854
		Δ*HA*	13,735.99	15,562.02	18,443.68	15,810.09
		Δ*HL*	−594,946	−61,614.7	−324,826	−209,042.6
		Δ*HH*	506,950.4	190,354.7	553,928.6	314,733.4
		*TL*	287.8609	286.2315	287.8129	287.5896
		*TH*	333.1618	309.4937	308.1277	308.7199
		*χ* ^2^	0.0049	0.0001	0.001	0.00002
		*r^2^*	0.98	0.99	0.98	0.99
Maize	SSI	*Ρ_Φ_*	0.5172	0.0381	0.205	0.0262
		*Τ_Φ_*	307.449	295.7491	309.8748	296.7376
		Δ*HA*	14,195.38	18,229.1	1781.998	16,834.68
		Δ*HL*	−553,757.2	−54,196.12	−37,369.65	−55,647.81
		Δ*HH*	891,681.9	58,529.74	28,065.65	50,060.75
		*TL*	287.8202	282.8056	295.9896	285.2618
		*TH*	308.3621	309.0867	328.3202	310.2237
		*χ* ^2^	0.003	0.0004	0.0024	0.0006
		*r^2^*	0.99	0.99	0.98	0.99
Potato	SSI	*Ρ_Φ_*	0.5145	0.0464	0.0784	0.0244
		*Τ_Φ_*	306.7806	296.2396	295.5563	295.4299
		Δ*HA*	15,762.87	9203.354	25,257.17	14,400.88
		Δ*HL*	−598,292.7	−123,805.7	−46,475.31	−109,684.6
		Δ*HH*	413,041	62,760.16	64,174.94	56,188.48
		*TL*	287.6528	287.6677	281.6108	287.0287
		*TH*	339.274	312.6241	307.4918	311.0272
		*χ* ^2^	0.0033	0.0001	0.0023	0.00002
		*r^2^*	0.99	0.99	0.99	0.99
Green pea	SSI	*Ρ_Φ_*	0.5261	0.0868	0.1352	0.0544
		*Τ_Φ_*	306.7962	306.9769	299.6282	306.2005
		Δ*HA*	16,243.73	8444.23	21,841.81	11,446.02
		Δ*HL*	−598,362	−60,000	−191095.7	−598,991.3
		Δ*HH*	585,099.1	110,672.7	310,905.3	709,999.9
		*TL*	287.6255	288.1208	286.7044	288.0605
		*TH*	329.2338	469.0172	308.4621	323.4305
		*χ* ^2^	0.0039	0.0012	0.0006	0.0002
		*r^2^*	0.99	0.98	0.99	0.99
Artificial diet	SSI	*Ρ_Φ_*	0.5221	0.1313	0.1432	0.0422
		*Τ_Φ_*	304.1798	311.8094	287.9168	299.8823
		Δ*HA*	18,298.26	1882.327	15,493.79	15,133.16
		Δ*HL*	−154894.7	−36,901.88	−70,880.84	−62,549.37
		Δ*HH*	779,997.6	30,264.05	119,927.5	85,965.03
		*TL*	286.4242	296.2487	287.9168	287.6329
		*TH*	308.3621	331.7074	309.0019	310.1774
		*χ* ^2^	0.0009	0.0001	0.00004	0.00006
		*r* ^2^	0.99	0.99	0.99	0.99

**Table 5 insects-13-00747-t005:** Estimated annual voltinism over a four-year period in Miryang, Korea, based on a biofix: 1 January, and date of spring emergence of *Spodoptera exigua* adults.

Year	Hosts	Biofix of 1 January		
		Thermal Constant (DD)	No. of Generations	Emergence Date
2018	Soybean	399.6	5.24	5 June
	Maize	463.0	4.70	8 June
	Potato	449.0	5.09	5 June
	Green pea	445.6	5.33	2 June
	Artificial diet	357.1	5.64	3 June
2019	Soybean	402.8	5.10	10 June
	Maize	463.1	4.58	14 June
	Potato	449.3	4.96	10 June
	Green pea	445.0	5.21	6 June
	Artificial diet	357.5	5.52	7 June
2020	Soybean	402.2	4.97	6 June
	Maize	446.1	4.46	9 June
	Potato	446.7	4.83	6 June
	Green pea	453.3	5.08	4-June
	Artificial diet	359.3	5.38	4 June
2021	Soybean	397.4	5.35	9 June
	Maize	471.9	4.80	13 June
	Potato	449.5	5.19	9 June
	Green pea	443.4	5.45	6 June
	Artificial diet	362.1	5.79	8 June

Spring emergence dates for *S. exigua* adults were predicted from the degree-day calculation from the same weather data of each year for all plant hosts (soybean, maize, potato and green pea), based on the lower development threshold.

**Table 6 insects-13-00747-t006:** ANOVA results for the combined effects of temperature and plant hosts (soybean, maize, potato and green pea), and interaction of temperature and host (plant hosts [soybean, maize, potato and green pea] and the artificial diet) on the development of eggs, larvae, pupae, and total immatures of *Spodoptera exigua*.

Stage	Source	DF	Sum of Squares	Mean Square	F Value	*p* > F
Egg	Model	29	15,967.546	550.605	1142.26	<0.0001
	Temperature	5	15,909.489	3181.898	6601.04	<0.0001
	Host	4	26.577	6.644	13.78	<0.0001
	Temperature x Host	20	31.478	1.574	3.27	<0.0001
Larva	Model	29	335,589.944	11,572.067	2146.31	<0.0001
	Temperature	5	321,971.811	64,394.362	11,943.40	<0.0001
	Host	4	3142.313	785.578	145.70	<0.0001
	Temperature x Host	20	10,475.821	523.791	97.15	<0.0001
Pupa	Model	29	146,514.971	5052.240	3002.86	<0.0001
	Temperature	5	145,298.478	29,059.696	17272	<0.0001
	Host	4	336.495	84.124	50.00	<0.0001
	Temperature x Host	20	879.998	43.999	26.15	<0.0001
Total immature	Model	29	1,039,677.490	35,850.948	4599.23	<0.0001
	Temperature	5	1,026,179.445	205,235.889	26,329.20	<0.0001
	Host	4	4851.001	1212.750	155.58	<0.0001
	Temperature x Host	20	8647.044	432.352	55.47	<0.0001

Overall model of seven temperatures (15, 20, 25, 27, 30, 35, and 40 °C), and host (plant hosts [soybean, maize, potato, green pea] and the artificial diet).

**Table 7 insects-13-00747-t007:** Adult longevity (day, mean ± SE) and sex ratio (female) of *Spodoptera exigua* reared on different plant hosts (soybean, maize, potato and green pea) and the artificial diet as a food source at constant temperatures.

Parameter	Hosts	Sex	Temperature (°C) (Mean ± SE)						
			40	35	30	27	25	20	15
Adult longevity	Soybean	F	-	9.11 ± 0.4ab	12.28 ± 0.3a *	10.09 ± 0.3ab *	10.26 ± 0.3ab *	8.80 ± 0.4b	8.91 ± 0.3b *
		M	-	8.42 ± 0.3b	9.31 ± 0.4ab	8.92 ± 0.1ab	9.46 ± 0.3ab	8.45 ± 0.3b	10.20 ± 0.3a
	Maize	F	-	8.02 ± 0.2bc *	9.26 ± 0.2a	8.45 ± 0.2b *	8.15 ± 0.3bc	7.50 ± 0.4c	7.85 ± 0.2bc
		M	-	6.96 ± 0.2bc	9.25 ± 0.4a	7.29 ± 0.2bc	8.06 ± 0.2b	6.91 ± 0.4c	8.04 ± 0.2b
	Potato	F	-	8.68 ± 0.1b *	8.83 ± 0.1b *	9.39 ± 0.1c *	9.52 ± 0.2a *	9.55 ± 0.2a *	7.70 ± 0.3c
		M	-	6.89 ± 0.2b	7.95 ± 0.2a	8.29 ± 0.2a	8.50 ± 0.2a	8.16 ± 0.3a	8.24 ± 0.3a
	Green pea	F	-	9.50 ± 0.3a *	8.68 ± 0.1c *	8.80 ± 0.2c	9.14 ± 0.2ab *	9.06 ± 0.2ab *	9.28 ± 0.2ab *
		M	-	8.05 ± 0.2bc	7.42 ± 0.2c	8.42 ± 0.2bc	8.62 ± 0.2b	7.98 ± 0.2bc	10.25 ± 0.2a
	Artificial diet	F	-	9.90 ± 0.2c *	11.76 ± 0.2a *	11.97 ± 0.3a *	10.25 ± 0.2c	11.25 ± 0.4ab *	10.68 ± 0.4bc *
		M	-	7.03 ± 0.4d	8.95 ± 0.5c	10.14 ± 0.3b	10.57 ± 0.6ab	9.92 ± 0.2bc	11.64 ± 0.2a
Sex ratio	Soybean	-	-	0.6	0.6	0.6	0.5	0.5	0.5
	Maize	-	-	0.5	0.5	0.6	0.4	0.5	0.5
	Potato	-	-	0.6	0.6	0.6	0.7	0.6	0.4
	Green pea	-	-	0.6	0.6	0.6	0.5	0.5	0.5
	Artificial diet	-	-	0.6	0.7	0.6	0.5	0.5	0.6

F-female and M-male; ANOVA for Soybean: Female-F_5, 93_ =13.92, *p* < 0.0001, Male-F_5, 74_ =5.20, *p* = 0.0004; Maize: Female-F_5, 84_ =7.78, *p* < 0.0001, Male-F_5, 92_ =8.38, *p* < 0.0001; Potato: Female-F_5, 98_ =16.86, *p* < 0.0001, Male-F_5, 102_ =7.00, *p* < 0.0001; Green pea: Female-F_5, 136_ =2.39, *p* = 0.0401, Male-F_5, 109_ =26.48, *p* < 0.0001; Artificial diet: Female-F_5, 94_ =9.06, *p* < 0.0001, Male-F_5, 67_ =15.48, *p* < 0.0001. Means followed by the same letters in a row are not significantly different among temperatures (ANOVA, Tukey’s HSD test, *p*< 0.05). *t*-test for Soybean: 35 °C, t = 1.25, *p* = 0.22; 30 °C, t = 6.29, *p* < 0.0001; 27 °C, t = 2.67, *p* = 0.01; 25 °C, t = 2.12, *p* = 0.04; 20 °C, t = 0.80, *p* = 0.43; 15 °C, t = 3.97, *p* = 0.001. Maize: 35 °C, t = 2.15, *p* = 0.0.03; 30 °C, t = 0.03, *p* = 0.97; 27 °C, t = 4.83, *p* < 0.0001; 25 °C, t = 0.42, *p* = 0.67; 20 °C, t = 1.16, *p* = 0.25; 15 °C, t = 0.66, *p* = 0.51. Potato: 35 °C, t = 8.42, *p* < 0.0001; 30 °C, t = 4.69, *p* < 0.0001; 27 °C, t = 4.92, *p* < 0.0001; 25 °C, t = 3.56, *p* = 0.001; 20 °C, t = 4.55, *p* < 0.0001, 15 °C, t = 1.27, *p* = 0.21. Green pea: 35 °C, t = 3.54, *p* = 0.0010; 30 °C, t = 5.94, *p* < 0.0001; 27 °C, t = 1.53, *p* = 0.1330; 25 °C, t = 2.90, *p* = 0.04; 20 °C, t = 4.19, *p* < 0.0001, 15 °C, t = 3.15, *p* = 0.003. Artificial diet: 35 °C, t = 7.25, *p* < 0.0001; 30 °C, t = 6.31, *p* < 0.0001; 27 °C, t = 4.27, *p* = 0.0002; 25 °C, t = 0.56, *p* = 0.58; 20 °C, t = 3.00, *p* = 0.005, 15 °C, t = 2.13, *p* = 0.04. ***** Significant difference between female and males (*t*-test, *p* < 0.005).

**Table 8 insects-13-00747-t008:** Composition of lipid, protein, and CHO (stachyose, raffinose, sucrose, glucose, galactose, and fructose) (mean ± SE) in soybean, maize, potato and green pea leaves.

Hosts	Lipid (%)	Protein (%)	CHO (mg/g)					
			Stachyose	Raffinose	Sucrose	Glucose	Galactose	Fructose
Soybean	4.77 ± 0.2a	47.54 ± 0.3a	n.a	n.a	9.28 ± 1.1c	25.29 ± 1.1b	n.a	0.26 ± 0.0b
Maize	4.74 ± 0.2a	29.78 ± 0.6c	n.a	n.a	20.31 ± 0.6b	8.54 ± 0.3c	n.a	0.61 ± 0.2a
Potato	3.58 ± 0.2b	24.52 ± 0.9d	n.a	1.28 ± 0.2b	10.38 ± 0.2c	6.62 ± 0.2c	n.a	0.53 ± 0.0ab
Green pea	5.31 ± 0.2a	35.94 ± 0.1b	n.a	2.58 ± 0.4a	60.49 ± 0.1a	34.68 ± 0.1a	0.24 ± 0.0	0.56 ± 0.1ab

n.a: not available. Means followed by the same letters in a column are not significantly different among plant hosts (soybean, maize, potato and green pea) (ANOVA, Tukey’s HSD test, *p* < 0.05).

## Data Availability

Data are available upon request to the corresponding author of this MS.

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
