# Peer review of "Interactive Effects of Temperature and Plant Host on the Development Parameters of Spodoptera exigua (Hübner) (Lepidoptera: Noctuidae)"

_insects, 2022, doi:10.3390/insects13080747_

Round 1
Reviewer 1 Report
The authors have graphed and presented their results clearly, drawing some attention to the implications of their findings. I found the study of interest and a good contribution to the knowledge of bioecology of invasive arthropods. The methods used for development data are, however, inappropriate for the objectives of the work (see my comments below). The resulting figures and tables are therefore insufficient, and of poor quality not helping to follow the reasoning throughout the manuscript.
The next draft of this paper will need to be dramatically different to have a chance at publication.
Firstly, the authors should explain about how they arrived at using solely the SSI function for their datasets. The authors conclude that the SSI model is the good-fitting model of the study, when that is clearly false from even a casual glance at Figs. 2-4 and results section, for their data sets. Optimal temperatures of 40Cs or 50Cs for immatures and eggs and curves not plunging down to hit the x axes are hard to comprehend. Statements like those may encourage other scientists to retain this model in future studies, when in fact the model should be relegated to the dustbin of history for your datasets.
It might be worth adding a brief statement about how you arrived at using solely the SSI function for their datasets??? Studies on the effects of temperature on insect developmental characteristics have been criticized because analyses commonly use models that are considered standard to the field of investigation or are subjected to biased selection or preferred for a particular taxonomic group (see Ann. Entomol. Soc. Am. 2015, 109: 211-215; Ann. Entomol. Soc. Am. 2017, 110: 302-309; and J. Econ. Entomol. 2020, 113: 633-645). As a result, alternative models that could provide superior fits to experimental datasets may be overlooked. Thus, it is valuable to further compare all other existing temperature-dependent growth models to the SSI function to find a better model than the SSI function. Then we can obtain more new candidate models by using the integral form for the developmental or growth equation or for estimated of population dynamics. A short note that was made by the authors in the introduction stating that other models do exist aside from SSI does not suffice.
Also, the Intro and Discussion provide no insight on the value of rearing arthropods in microcosms at fluctuating temperatures or quality control restrictions that have been raised by these other researchers and how do these results compare to those of insects reared at constant temperatures. Generally, temperature-driven development rate models for pest and associated natural enemy species have been constructed from studies performed at constant temperatures including this study. In the field, however, ambient temperatures fluctuate over time. Exposure to thermal variation during development significantly affects development times and subsequent adult fitness (see J. Econ. Entomol. 2019, 112: 1560-1574; and J. Econ. Entomol. 2019, 112:1062-1072). Failing to consider the effect of these temperature fluctuations on insect development and longevity could result in inaccurate predictions generated by models using (less realistic) constant temperature data. Consequently, the parameter estimates (e.g., rates of development, upper and lower thresholds) and the goodness of fit of the models (i.e., how accurately they predict development times over a range of temperatures) could differ appreciably between models fit to constant versus fluctuating temperature datasets. Consequently, phenology predictions using models fit to constant temperature data could erroneously predict shorter- or longer-than-actual generation times for insects during the season (see J. Econ. Entomol. 2019, 112:1062-1072). This is a critical limitation of the study, and the authors must concede and discuss this.
My main concern is that the authors are extrapolating the applicability of their results beyond what the design supports (i.e., the models in this study were developed from seven sets of highly artificial constant temperatures). Some of the authors statement would be much stronger if they tie and compare their work to the body of literature that has built up from rearing insects in microcosms at more realistic fluctuating temperatures. Adding these details will improve the discussion. This is not to diminish the data gathered in this study, they are of value. But it is important for the authors not to overgeneralize, and to warn the reader (including regulatory agencies…) against doing so as well.
Best of luck to you!
Author Response
The authors have graphed and presented their results clearly, drawing some attention to the implications of their findings. I found the study of interest and a good contribution to the knowledge of bioecology of invasive arthropods. The methods used for development data are, however, inappropriate for the objectives of the work (see my comments below). The resulting figures and tables are therefore insufficient, and of poor quality not helping to follow the reasoning throughout the manuscript.
The next draft of this paper will need to be dramatically different to have a chance at publication.
-->Authors thanks and deeply appreciated for your important comments and suggestions. We have taken your suggestion as valuable inputs to improve the MS. Based on the comments and suggestions, we have amended accordingly and information added.
Firstly, the authors should explain about how they arrived at using solely the SSI function for their datasets. The authors conclude that the SSI model is the good-fitting model of the study, when that is clearly false from even a casual glance at Figs. 2-4 and results section, for their data sets. Optimal temperatures of 40Cs or 50Cs for immatures and eggs and curves not plunging down to hit the x axes are hard to comprehend. Statements like those may encourage other scientists to retain this model in future studies, when in fact the model should be relegated to the dustbin of history for your datasets.
It might be worth adding a brief statement about how you arrived at using solely the SSI function for their datasets??? Studies on the effects of temperature on insect developmental characteristics have been criticized because analyses commonly use models that are considered standard to the field of investigation or are subjected to biased selection or preferred for a particular taxonomic group (see Ann. Entomol. Soc. Am. 2015, 109: 211-215; Ann. Entomol. Soc. Am. 2017, 110: 302-309; and J. Econ. Entomol. 2020, 113: 633-645). As a result, alternative models that could provide superior fits to experimental datasets may be overlooked. Thus, it is valuable to further compare all other existing temperature-dependent growth models to the SSI function to find a better model than the SSI function. Then we can obtain more new candidate models by using the integral form for the developmental or growth equation or for estimated of population dynamics. A short note that was made by the authors in the introduction stating that other models do exist aside from SSI does not suffice.
-->The authors appreciate your valuable comments on the manuscript. The authors understand there are several temperature-dependent development models in the literatures. Based on observed data, we analyzed the data using different models and then selected the SSI model. Quinn (2021) suggested that “the SSI function performed as well as or relatively better than other functions of comparable or lower complexity in terms of R2, AICC-based rankings, ΔAICC values, and prediction errors, which may recommend its more widespread use in future studies. For your kind information, we added information related to model in the discussion section along with references. Pls see LN 681-698.
Also, the Intro and Discussion provide no insight on the value of rearing arthropods in microcosms at fluctuating temperatures or quality control restrictions that have been raised by these other researchers and how do these results compare to those of insects reared at constant temperatures. Generally, temperature-driven development rate models for pest and associated natural enemy species have been constructed from studies performed at constant temperatures including this study. In the field, however, ambient temperatures fluctuate over time. Exposure to thermal variation during development significantly affects development times and subsequent adult fitness (see J. Econ. Entomol. 2019, 112: 1560-1574; and J. Econ. Entomol. 2019, 112:1062-1072). Failing to consider the effect of these temperature fluctuations on insect development and longevity could result in inaccurate predictions generated by models using (less realistic) constant temperature data. Consequently, the parameter estimates (e.g., rates of development, upper and lower thresholds) and the goodness of fit of the models (i.e., how accurately they predict development times over a range of temperatures) could differ appreciably between models fit to constant versus fluctuating temperature datasets. Consequently, phenology predictions using models fit to constant temperature data could erroneously predict shorter- or longer-than-actual generation times for insects during the season (see J. Econ. Entomol. 2019, 112:1062-1072). This is a critical limitation of the study, and the authors must concede and discuss this.
My main concern is that the authors are extrapolating the applicability of their results beyond what the design supports (i.e., the models in this study were developed from seven sets of highly artificial constant temperatures). Some of the authors statement would be much stronger if they tie and compare their work to the body of literature that has built up from rearing insects in microcosms at more realistic fluctuating temperatures. Adding these details will improve the discussion. This is not to diminish the data gathered in this study, they are of value. But it is important for the authors not to overgeneralize, and to warn the reader (including regulatory agencies…) against doing so as well.
--> Fluctuating temperatures may be more resource-consuming environments than constant temperature held at equivalent mean temperature and tend to produce divergent responses. There are also other factors such as relative humidity, rearing density, sex and all the interactions between these factors influencing on development rate in insects. Future studies may be needed to examine the biological performance of S. exigua under different environmental conditions affecting insect development and fecundity. Your input will be incorporated in the further research studies. In addition, information added in the discussion section along with references. Pls see LN 716-727.
Once again we deeply appreciated reviewers’ comments and suggestion. This is a huge input for the improvement of MS. Thanks
Reviewer 2 Report
Reviewer # 1E
Manuscript ID: insects-1841892
· The manuscript ID: insects-1841892 ¨ Interactive effects of temperature and crop host on the development parameters of Spodoptera exigua (Hübner) (Lepidoptera: Noctuidae) ¨, has relevant information on S. exigua and environmental factors that can be useful to the body of knowledge of the species in the world. However, there are some errors (also some speculation) that must be removed from the writing, as well as substantially reducing the repetition of some analyzes that seem unnecessary.
· It is generally well written in English, although the use of words or adjectives that are more commonly used in the area needs to be improved. For example, plant host instead of crop host. Sometimes authors use ¨development time or developmental time¨ as synonyms, it is recommended to use the correct adjectives throughout the manuscript (noted in the pdf file).
· Basically two objectives were established. The first was to know the effect of temperature and host plant on developmental time, and to establish a model to explain this. That was the only valid aim of the study. The second pseudo-goal made no sense. It is absurd to think that establishing the nutritional quality of the seeds (of host plants) can be used to explain nutrition that larvae could have obtained from the leaves (lines 98-100; 149-153). It would have been good to do a bromatological analysis of the leaves, but if they didn't, then everything related to it should be removed from the manuscript. No inference should be made about the nutritional quality of the host plant if it was not measured by what the larvae were fed.
· In general the results are well written. Nevertheless, it seems that the way the authors wanted to describe each analysis is excessive. I think, they can summarize and eliminate some images or pictures; 36 pages could surely be reduced to 25 or 27 and nothing valuable would be lost. Only unnecessary information would be removed, but if they want to keep some of that, authors can send it to the section of data that is shared but does not appear in the article (additional data).
· The discussion should also be slightly redirected and avoid repetition, and speculation in the case of the nutritional value of the host plant and its effect on development. Authors did not evaluate the nutritional value of the foliage (leaves) but of the seeds, and it does not make sense to explain the nutrient effect of the seeds when larvae were not fed on them.
· Some additional comments were included in the pdf file; I hope that those comments would help to improve the manuscript.

Author Response
The manuscript ID: insects-1841892 ¨ Interactive effects of temperature and crop host on the development parameters of Spodoptera exigua (Hübner) (Lepidoptera: Noctuidae) ¨, has relevant information on S. exigua and environmental factors that can be useful to the body of knowledge of the species in the world. However, there are some errors (also some speculation) that must be removed from the writing, as well as substantially reducing the repetition of some analyzes that seem unnecessary.
--> Thanks and we deeply appreciated for your important comments and suggestions. Most of the comments/suggestions were respected, and we have edited MS, and amended accordingly based on the comments and suggestions made on pdf.
It is generally well written in English, although the use of words or adjectives that are more commonly used in the area needs to be improved. For example, plant host instead of crop host. Sometimes authors use ¨development time or developmental time¨ as synonyms, it is recommended to use the correct adjectives throughout the manuscript (noted in the pdf file).
-->Thanks for valuable comment. Words/adjectives that were used in MS were corrected/changed throughout MS based on the comments made in pdf. Further, MS was once again edited with professional English editing service for better information delivery. Pls see MS (from title to table and graphs).
Basically two objectives were established. The first was to know the effect of temperature and host plant on developmental time, and to establish a model to explain this. That was the only valid aim of the study. The second pseudo-goal made no sense.
--> Thanks for valuable comment. Agree with you, two objectives are the major one. Information relative to objectives was changed, and removed pseudo-goal. Pls see LN 95-101.
It is absurd to think that establishing the nutritional quality of the seeds (of host plants) can be used to explain nutrition that larvae could have obtained from the leaves (lines 98-100; 149-153). It would have been good to do a bromatological analysis of the leaves, but if they didn't, then everything related to it should be removed from the manuscript. No inference should be made about the nutritional quality of the host plant if it was not measured by what the larvae were fed.
--> Thanks for important comment. First of all, we would like to inform you that in the first MS, we had mistakenly written seed (in LN 220) instead of plant leaves. Obviously there is no meaning to perform nutrient analysis with seeds as larvae were fed with leaves of each host plant in the experiment. So, in the nutrient contents analysis, we used the leaves rather than seeds of plant hosts during study period. Words were changed in M&M section. Pls see LN 223 to 241.
So far bromatological analysis, it would be great, if we did perform bromatological analysis for assessing components other than plant nutrients (lipid, protein, and CHO) such as chemical substances and defense materials associated with used plant hosts. However, we are not able to perform bromatological analysis at least in this moment due to some technical difficulties and resources constraints, but it will definitely be considered and performed in further studies. In the present study, basically we focused on the plant nutrient contents because those plant nutrient contents are critical and play a vital role in the development of insects though some effects are also given by other substances associated with plant hosts. So, we believe that information presented in the MS carry meaning and helps to deliver relevant information for clear understanding the study.
Reviewer comment in pdf (LN 125-128),
-->Each plant host are different plant species with different growth characteristics and growth timing at same environmental and soil conditions. Some of the plant nutrients and morphological characteristics (ex. trichrome on leaf, growth pattern) do vary according to growth stage of each plant host. To minimize those variations, planting time of plants were synchronized in the study.
In general the results are well written. Nevertheless, it seems that the way the authors wanted to describe each analysis is excessive. I think, they can summarize and eliminate some images or pictures; 36 pages could surely be reduced to 25 or 27 and nothing valuable would be lost. Only unnecessary information would be removed, but if they want to keep some of that, authors can send it to the section of data that is shared but does not appear in the article (additional data).
--> Thanks for your concerns and suggestion. Some of the figures can be merged and kept in the addition data section which definitely will minimize the pages of MS. But we believe that clear presentation of information and figures/tables in the main body of MS makes more clearness and easy readable, and disseminate clear information to the readers at a glance.
The discussion should also be slightly redirected and avoid repetition, and speculation in the case of the nutritional value of the host plant and its effect on development. Authors did not evaluate the nutritional value of the foliage (leaves) but of the seeds, and it does not make sense to explain the nutrient effect of the seeds when larvae were not fed on them.
-->Thanks once again for your comments and suggestion. As suggested in the pdf, some parts of discussion were changed. In case of nutrient content analysis, as we mentioned above comment section, we did nutrient analysis with leaves of plant hosts, not with seeds, as experiment was carried out by feeding leaves of host plant to the larvae of S. exigua. So, we believe that information related to plant nutrient contents provided in the MS carry meaning. Further, information on ‘seed’ has changed/replaced with ‘leave’ in the M&M section. Pls see LN 223 to 241.
Some additional comments were included in the pdf file; I hope that those comments would help to improve the manuscript.
--> Thanks for your valuable time and efforts to make comments and suggestion regarding MS. Defiantly, your contribution helped a lot to improve MS. Appreciated.
Round 2
Reviewer 1 Report
Authors have done a fine job addressing all of my previous concerns and those of other reviewers. I have no additional suggestions for the paper. Thank you.
Author Response
Rievewer-1
Comments and Suggestions for Authors
Authors have done a fine job addressing all of my previous concerns and those of other reviewers. I have no additional suggestions for the paper. Thank you.
-->Thanks once again, and we deeply appreciated for your valuable time for reviewing the MS. Due to your suggestions and comments, we could improve MS and delivery information clearly. In addition, we also could learn things about MS writing. It will be guidance for us.
Thanks again for your contribution.
Reviewer 2 Report
Reviewer # 1E (second revision)
Manuscript ID: insects-1841892
· The authors attended almost all observations of the first review. The manuscript ID: insects-1841892 ¨ Interactive effects of temperature and crop host on the development parameters of Spodoptera exigua (Hübner) (Lepidoptera: Noctuidae) ¨, has improved considerably. I just marked a few minor comments in the pdf file. For example Lines 35, 50, 75, 95, 117, 135, 137, 142, 150, 176, 213, etc.
· Authors still has an error in line 221-222, they did not use seeds for the analysis (according to lines 213 and 214). · The titles of tables could be improve. I am using just as example Table 1: It is written: Table 1. Development time (day, mean ± SE) for Spodoptera exigua using plant hosts(soybean, maize, potato, green pea) and the artificial diet
as food sources at a constant temperature. It seems better: Table 1. Developmental time in days (mean ± SE) for Spodoptera exigua
on soybean, maize, potato, and green pea leaves, and an artificial diet at constant temperatures. · Line 525, Is there (raffinose) a difference? · Line 532, the title could be improve. · Line 606, it is written …tested crop hosts (Table 8). It could be better…host plant leaves. · Line 651-652, Table 9 and those lines seem completely unnecessary.

Author Response
Reviewer-2
The authors attended almost all observations of the first review. The manuscript ID: insects-1841892 ¨ Interactive effects of temperature and crop host on the development parameters of Spodoptera exigua (Hübner) (Lepidoptera: Noctuidae) ¨, has improved considerably. I just marked a few minor comments in the pdf file.
-->Thanks once again, and we deeply appreciated for your valuable time for reviewing the MS. Most of the comments/suggestions were respected, and we have edited MS, and amended accordingly based on the comments and suggestions made on pdf.
For example Lines 35, 50, 75, 95, 117, 135, 137, 142, 150, 176, 213, etc.
For LN 39: --> We added ‘approximately’ instead of ‘approx.’ and deleted ‘ and the artificial diet’. pls see LN 40-41.
For LN 50: --> We added information regarding instars. Pls see LN 51-52.
For LN 75: --> deleted ‘asterisk’.
For LN 95: --> deleted ‘das’ in between ‘host and soybean’.
For LN 117: --> Deleted word ‘test’. Pls see LN 122.
For LN 135, 137, 142, and 150: --> spacing were corrected.
For LN 176: -->Deleted ‘full stop’.
For LN 213: -->Thanks. Sentence was modified. Pls see LN 224-226.
For LN 221: -->Thanks. Word ‘seeds’ replaced with ‘leaves’. Pls see LN 234.
For LN 242-244: -->Managed the ‘double space’.
Authors still has an error in line 221-222, they did not use seeds for the analysis (according to lines 213 and 214).
-->Thanks. Word ‘seeds’ replaced with ‘leaves’. Pls see LN 234.
The titles of tables could be improve. I am using just as example Table 1: It is written: Table 1. Development time (day, mean ± SE) for Spodoptera exigua using plant hosts
(soybean, maize, potato, green pea) and the artificial diet
as food sources at a constant temperature. It seems better: Table 1. Developmental time in days (mean ± SE) for Spodoptera exigua on soybean, maize, potato, and green pea leaves, and an artificial diet at constant temperatures.
--> Thanks. Table titles have improved where necessary and some of the table titles were kept as it is, because it also provides clear information. Pls see table titles.·
Line 525, Is there (raffinose) a difference?
--> Thanks. Yes, there was a difference. Just looking by probability value (p=0.0529), it seems slightly over than p=0.05. Generally, in statistics, up to p=0.05 is considered as a significantly different level. Even in some cases, up to p=0.09 is considered as ‘suggestive significantly different’. So, we had written as significantly different for raffinose.
Line 532, the title could be improve.
--> Title of table modified as per suggestion. Pls see LN 554-556.
Line 606, it is written …tested crop hosts (Table 8). It could be better…host plant leaves.
--> Words ‘tested crop hosts’ replaced with ‘host plant leaves’ as suggested. Pls see LN 633.
Line 651-652, Table 9 and those lines seem completely unnecessary.
--> information above in the discussion section and Table 9 deleted as per suggestion. Pls see LN 679-680 and Table 9.
Round 3
Reviewer 2 Report
Hello,
I reviewed only my last comments from the second review,all of them were taken care of. I have no further comments. Good work